# Astrocytes integrate and drive action potential firing in inhibitory subnetworks

Tara Deemyad[1,2], Joel Lüthi [1,3] & Nelson Spruston [1]

Many brain functions depend on the ability of neural networks to temporally integrate transient inputs to produce sustained discharges. This can occur through cell-autonomous mechanisms in individual neurons and through reverberating activity in recurrently connected neural networks. We report a third mechanism involving temporal integration of neural activity by a network of astrocytes. Previously, we showed that some types of interneurons can generate long-lasting trains of action potentials (barrage firing) following repeated depolarizing stimuli. Here we show that calcium signaling in an astrocytic network correlates with barrage firing; that active depolarization of astrocyte networks by chemical or optogenetic stimulation enhances; and that chelating internal calcium, inhibiting release from internal stores, or inhibiting GABA transporters or metabotropic glutamate receptors inhibits barrage firing. Thus, networks of astrocytes influence the spatiotemporal dynamics of neural networks by directly integrating neural activity and driving barrages of action potentials in some populations of inhibitory interneurons.

[1] Janelia Research Campus, Howard Hughes Medical Institute, Ashburn, VA 20147, USA. [2]Present address: Department of Neurobiology, School of Medicine, University of Pittsburgh, Pittsburgh, PA 15213, USA. [3]Present address: Institute of Molecular Life Sciences, University of Zürich, Zürich 8057, Switzerland. Correspondence and requests for materials should be addressed to N.S. (email: sprustonn@janelia.hhmi.org)

A traditional view of the nervous system delineates a clear division of labor between neurons and glia. Neurons process information encoded in the form of synaptic potentials and action potentials, with excitatory neurons promoting action potential firing in their targets, inhibitory neurons limiting and controlling the timing of action potential firing, and other neurons modulating the function of excitatory and inhibitory neurons. Glial cells provide a support role for neurons by maintaining extracellular homeostasis (astrocytes), as well as insulating axons (Schwann cells and oligodendrocytes), and providing immune surveillance of the nervous system (microglia)[1]. A broader view, however, includes the possibility that glial cells are active participants in the information processing capabilities of neural circuits[2]. Indeed, there is ample evidence that astrocytes influence network states and even affect animal behavior[3–7]. Astrocytes can influence neuronal function in a variety of ways, most notably by regulating the concentration of ions and neurotransmitters[3,8] and by releasing gliotransmitters, which can act directly on neuronal receptors[9–13], collectively modulating neuronal excitability, synaptic transmission, and plasticity[14–16]. Importantly, the concept of the tripartite synapse[9,17–19] is central to our understanding of astrocyte function in the nervous system.

A specific proposal is that astrocytes can integrate neural signals on spatial and temporal scales that are different from, but complementary to, the function of neurons and produce persistent neural activity[20,21]. We have shown that specific subtypes of inhibitory neurons can generate high-frequency barrages of action potentials that outlast stimulation by tens of seconds[22]. This persistent neural activity has an abrupt onset but is triggered only following slow integration of hundreds of action potentials evoked by repeated stimulation of the neuron. Furthermore, the resultant barrages of action potentials are generated in the distal axon, far from the soma[22,23]. This unusual form of action potential integration and persistent neural activity (barrage firing) has been observed in genetically defined subsets of interneurons of several brain areas, both in vitro and in vivo[22–26]. Although the behavioral implications of interneuron barrage firing are unknown, insight into its underlying mechanisms may reveal a novel means by which temporal integration and persistent neural activity are achieved.

Several clues suggested to us that astrocytes might be directly involved in the generation of barrage firing[22,23]. First, although barrage firing is eliminated by inhibition of gap junctions, it could be induced in mice lacking the principal connexin subunit that forms electrical synapses between neurons. This suggests that the astrocytes, with their extensive gap junction connections, could underlie barrage firing. Second, decreasing the concentration of extracellular calcium or blocking L-type voltage-gated calcium (L-Ca$_v$) channels inhibits barrage firing, whereas buffering calcium in the recorded interneurons does not, thus suggesting that calcium plays a role elsewhere. Together, these lines of evidence hint at a possible role for astrocytes in the generation of barrage firing. Here we probe this idea more directly. Our results suggest that, following stimulation of interneurons, astrocytes depolarize, increase their internal calcium, and release glutamate, which acts on metabotropic glutamate receptors in axon terminals of interneurons to initiate barrage firing.

## Results

### Barrage firing in a population of inhibitory interneurons.
We performed patch-clamp recordings from interneurons near the border of stratum radiatum (SR) and stratum lacunosum-moleculare (SLM) in hippocampal area CA1 in a mouse line expressing green fluorescent protein (GFP) under the neuropeptide Y (NPY) promoter (hereafter NPY interneurons). Barrage firing can be induced in these neurons using any stimulation protocol that drives hundreds of spikes at rates above ~5 Hz, which is well within normal firing rates for hippocampal interneurons in vivo[22,23]. We induced barrage firing by injecting current steps to elicit action potential firing in NPY interneurons (Fig. 1a, b; 1 s, 180–700 pA current injections, followed by 2 s rest). Barrage firing began after hundreds of action potentials were evoked by repeated current injections that induced ~40 spikes during each 1-s step (Fig. 1c, $604 \pm 72$ evoked action potentials, barrage firing duration $33.2 \pm 13.2$ s, $n = 18$). Approximately 90% of NPY interneurons displayed barrage firing ($n = 108/121$), whereas it was never observed in GFP-negative interneurons in this region ($n = 0/27$). Spikes generated during barrage firing had unusual properties[22,23], such as a low apparent voltage threshold and the occasional appearance of spikelets (Fig. 1 d).

**Astrocytic depolarization and barrage firing.** As an initial test of whether astrocytes are activated during the induction of barrage firing, we recorded the membrane potential of an identified astrocyte, stained with the astrocyte-selective dye sulforhodamine 101 (SR101), while inducing barrage firing in a nearby NPY interneuron (50–100 μm from the astrocyte, Fig. 1b). We often observed a depolarization of astrocytes prior to the onset of NPY interneuron barrage firing (Fig. 1d, e). Astrocyte depolarization (peak: $10.3 \pm 1.8$ mV, $n = 14$) was observed in 80% of paired-recording experiments exhibiting barrage firing but never occurred in control experiments with weaker interneuron stimulation (Fig. 1d–h, $n = 10$).

A number of observations suggest a complex relationship between the depolarization of the single recorded astrocyte and the induction of barrage firing in the stimulated NPY interneuron. First, the amplitude of single-astrocyte depolarization was not correlated with the duration or firing frequency of the barrage firing ($R^2 < 0.0001$). Second, the main astrocyte depolarization occurred abruptly (20–80% rise time: $1.3 \pm 0.2$ s, $n = 14$) during the long stimulation time required to induce barrage firing ($1.6 \pm 0.1$ min from the start of stimulation, $n = 14$). Third, this abrupt component of astrocyte depolarization preceded the onset of barrage firing by tens of seconds ($29.5 \pm 5.8$ s, $n = 14$; Fig. 1i). Fourth, in most cases the astrocyte depolarization decayed prior to the onset of barrage firing in the NPY interneuron (Fig. 1h). Fifth, in some of recordings ($n = 4/14$), astrocyte depolarization outlasted barrage firing in the NPY interneuron (Fig. 1j). Together, these findings indicate a complex relationship between single-astrocyte depolarization and the onset of barrage firing in the NPY interneuron.

These complexities must be interpreted in the context of the fact that recordings were obtained from single astrocytes in a network generated by their gap junction connections. Because ~80% of the recorded astrocytes were depolarized, it seems likely that each NPY interneuron interacts with many but not all astrocytes in its vicinity. When single astrocytes were filled with biocytin, in a subset of slices ($n = 10$) post hoc staining revealed that the electrode contents spread through a network of astrocytes ($58 \pm 8$ astrocytes per slice), as reported previously[27]. Thus several astrocytes connected by gap junctions likely depolarize prior to the generation of barrage firing.

We have previously shown that the neuronal gap junction subunit Cx36 is not involved in induction of barrage of firing[23]. To test whether gap junctions in astrocytes are involved, we recorded from NPY cells in three lines of mice lacking one or both of the two main connexins in astrocytes: Cx30KO, Cx43floxKO, and Cx43flox/Cx30 KO (each crossed with NPY-GFP mice, as well as GFAP-Cre mice for the flox variants).

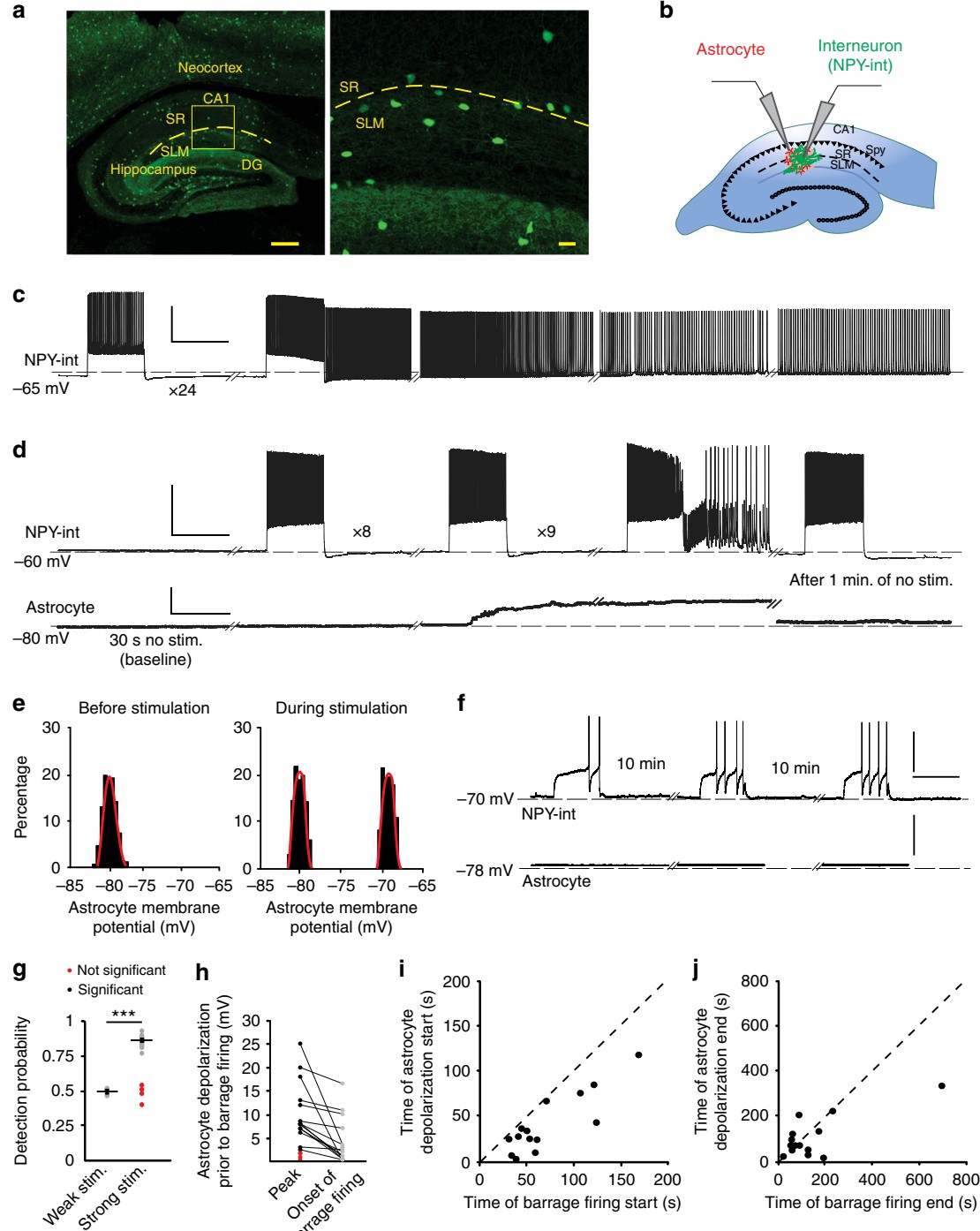

The probability of barrage firing decreased in NPY cells in all of these knockout (KO) mice; it was generated in 45% of cells in Cx30KO mice (9/20 cells in 2 animals), 60% in Cx43KO mice (12/20 cells in 2 animals), and 26% in Cx43flox/Cx30KO mice (9/34 cells in 2 animals; Supplementary Fig. 1a–b). These lowered percentages (vs. ~90% in control mice) strongly suggest a role for gap junctions between astrocytes in the generation of barrage firing.

To test whether the observed correlation between astrocyte depolarization and barrage firing is causal, we depolarized astrocytes by bath-applying 30 μM barium chloride (BaCl$_2$), which has been shown to block inwardly rectifying potassium (Kir) channels[28]. Kir channels are required for efficient potassium

buffering by astrocytes[28,29], and it has been suggested that Kir channels are abundant in astrocytes, as evidenced by the significant depolarization of astrocytes (up to 50 mV) in a Kir4.1 conditional knockout mouse model[30].

Barrage firing could be induced repeatedly in the same NPY interneuron, allowing it to be examined before and after drug application in the same cells. This is important, because the properties of barrage firing vary significantly across cells. In a series of control experiments, we found that the number of action potentials required to induce barrage firing varied from 160 to 1600, and the duration varied from less than few seconds to >1 min (Supplementary Fig. 2). As a further control, in the same set of NPY interneurons in SLM/SR, we tested whether changes in

**Fig. 1** Induction of barrage firing is correlated with astrocyte depolarization. **a** Left, GFP-labeled interneurons in NPY-GFP mouse line are present throughout the neocortex, hippocampus, and dentate gyrus (DG). Right, Labeled interneurons near the border of stratum lacunosum-moleculare (SLM) and stratum radiatum (SR) layers of area CA1. Scale bars, 300 μm left and 20 μm right panel. **b** Schematic of simultaneous recording from an NPY interneuron (NPY-int) and an astrocyte. **c** Example of barrage firing induction in an NPY interneuron with repeated depolarization by 1-s current injections followed by 2 s of rest. Scale bars, 40 mV, 1 s. Dashed lines represent reference potentials near the cell's resting potential (value indicated at left). **d** Example of an astrocyte depolarizing by ~15 mV (20–80% rise time of 1.3 s) before induction of barrage firing in a nearby NPY interneuron. The membrane potential of the astrocyte returned to resting potential 72 s after barrage firing ended. Scale bars, 40 mV for NPY-int, 15 mV for astrocyte,1 s. **e** All-points histograms of astrocyte membrane potential during baseline recording and during a stimulation trial before barrage firing; same recording as in **d**. **f** Weak depolarization of NPY interneurons did not result in barrage firing or a change in membrane potential of a nearby astrocyte. Scale bars, 40 mV, 1 s. **g** Detection probability for a change in astrocytic membrane potential was high prior to the onset of interneuron barrage firing (strong; $n = 18$ from 12 mice) but not in trials with weak stimuli ($n = 10$ from 6 mice) (i.e., ~50% or chance detection probability; ***$p < 0.0001$). In some recordings with strong interneuron stimulation and barrage firing, the detection probability of astrocyte membrane potential change was low (red; $n = 4/18$). Error bars represent SEM. **h** Astrocytes depolarized significantly prior to barrage firing in the majority of double recordings using strong stimulation (black; $n = 14/18$ cells from 9 mice). In a few cases, astrocytic depolarization was not observed prior to barrage firing (red; $n = 4/18$ from 3 mice). **i** Onset of astrocyte depolarization vs. barrage firing onset time. Note that all astrocytes depolarized prior to barrage firing generation (i.e., all points fall below unity line; $n = 14$ from 9 mice). **j** Return of astrocyte membrane potential to resting potential was not correlated with end of barrage firing ($n = 14$ from 9 mice). Data are represented as mean ± SEM

the properties of barrage firing were observed from trial to trial. When barrage firing was induced five times in the same neurons (one trial every 5 min), we found no significant difference in the number of action potentials required for generation of barrage firing across the five epochs (repeated-measures analysis of variance (ANOVA), $p = 0.53$, $n = 23$, Supplementary Fig. 2a–c). Thus, the effects of drugs (or other manipulations) on the induction of barrage firing can be interpreted readily.

In paired recordings from NPY interneurons and nearby astrocytes, we found that blocking Kir channels with 30 μM $BaCl_2$ resulted in a reversible depolarization of astrocytes (13.5 ± 2.6 mV, $n = 6$) but not of NPY interneurons (0.9 ± 0.2 mV, $n = 6$; Fig. 2a, b). In the presence of $BaCl_2$, fewer evoked action potentials were required to induce barrage firing in NPY interneurons compared to control conditions (602 ± 109 spikes after $BaCl_2$ vs. 1230 ± 177 spikes in control, $n = 6$; paired $t$ test, $p = 0.002$, Fig. 2c, d). This gain-of-function finding supports the notion that astrocytes play a causal role in the initiation of barrage firing. However, $BaCl_2$-evoked astrocyte depolarization did not affect the duration of action potentials during barrage firing (Fig. 2e). Furthermore, the time between depolarization of astrocytes and induction of barrage firing was not affected by $BaCl_2$ application (32.7 ± 7.8 s after $BaCl_2$, vs. 31.2 ± 7.2 s in control, $n = 5$; paired $t$ test, $p = 0.8$).

These results suggest that $BaCl_2$ facilitates barrage firing by blocking Kir channels in astrocytes, thus depolarizing them and facilitating barrage firing, either by lowering the threshold or accelerating the process that causes astrocytes to trigger sustained firing in NPY interneurons. Because $BaCl_2$ was applied to the bath, it acts on a network of astrocytes. Although we cannot rule out effects on other cell types, bath application of $BaCl_2$ did not result in barrage firing in pyramidal cells ($n = 4$) or non-NPY interneurons ($n = 5$), suggesting a specific relation between astrocytes and NPY interneurons.

**Astrocytic calcium and barrage firing**. We next asked whether an increase in astrocytic calcium signaling and the expected signaling to neighboring neurons via gliotransmitter release[31,32] is associated with barrage firing. We expressed a genetically encoded calcium indicator (GCaMP3) in astrocytes using viral transfection under the control of a glial fibrillary acidic protein (GFAP) promoter and imaged calcium-dependent fluorescence changes in astrocytes of hippocampal slices (Fig. 3a). We averaged the activity of all astrocytes in the field of view (GCaMP3-expressing astrocytes in the 200 × 200 μm² imaged area) during induction of barrage firing with current injections into a nearby NPY interneuron. While only small fluctuations in

calcium-dependent fluorescence were observed at rest, calcium-dependent fluorescence increased in astrocytes in response to stimulation of a nearby NPY interneuron. The observed calcium fluctuations encompassed the somata and processes of multiple astrocytes (average $\Delta F/F = 1.7 ± 0.2$, $n = 12$ slices, 4 ± 0.5 somata/slice, 22 ± 1 processes; Fig. 3b, c). In all recordings, barrage firing always occurred during a high calcium signal, indicating increased astrocytic network activity (Fig. 3c and Supplementary Fig. 7). Weaker current steps, which did not induce barrage firing, were never associated with increases in calcium-dependent fluorescence ($n = 0/6$; Fig. 3d–f). As with astrocyte depolarization, increases in astrocytic calcium preceded the onset of barrage firing in NPY interneurons by several seconds (10.75 ± 2.3 s, $n = 12$; Fig. 3g). Furthermore, termination of astrocytic depolarization was not correlated with the termination of barrage firing (Fig. 1j) and calcium signaling outlasted barrage firing >90% of the time ($n = 11/12$; Fig. 3h). Together, these findings suggest that multiple astrocytes may be involved in the temporal integration of neural activity and initiation (not termination) of barrage firing.

To test whether the observed relationship between astrocytic calcium signaling and the initiation of barrage firing in NPY interneurons is causal, we buffered calcium in astrocytes. First, we recorded from an NPY interneuron and induced barrage firing (twice, with a 5-min interval in between). Next, we recorded from a nearby astrocyte using an electrode containing 50 mM BAPTA. Finally, after a waiting period of 20 min for BAPTA to spread to neighboring astrocytes[27,33], we repeated the barrage-firing induction protocol in the same NPY interneuron (Fig. 4a). The membrane potential of astrocytes was not affected by using BAPTA inside the patch pipette, and post hoc staining of biocytin-filled astrocytes confirmed that astrocytic gap junctions were not blocked when patching with BAPTA-containing pipettes (56.2 ± 5.1 astrocytes stained; $n = 9$). Chelating astrocytic calcium with BAPTA inhibited the induction of barrage firing in >90% of the experiments ($n = 24/26$). Interestingly, astrocytes were still depolarized in the presence of BAPTA in response to stimulation of NPY interneurons (10.6 ± 3.2 mV) but chelating calcium did not change the time from onset of stimulation of NPY neurons to depolarization of the astrocytes. In two cases, BAPTA did not affect generation of barrage firing. However, in the majority of cases ($n = 18/26$), barrage firing was only observed after evoking many more action potentials than under control conditions (1179 ± 124 spikes with BAPTA vs. 720 ± 78 spikes in control, $n = 18$, paired $t$ test, $p = 0.0003$, Fig. 4b, c). In other cases ($n = 6/26$), barrage firing could not be induced at all after astrocytic BAPTA loading, even after evoking thousands of action potentials

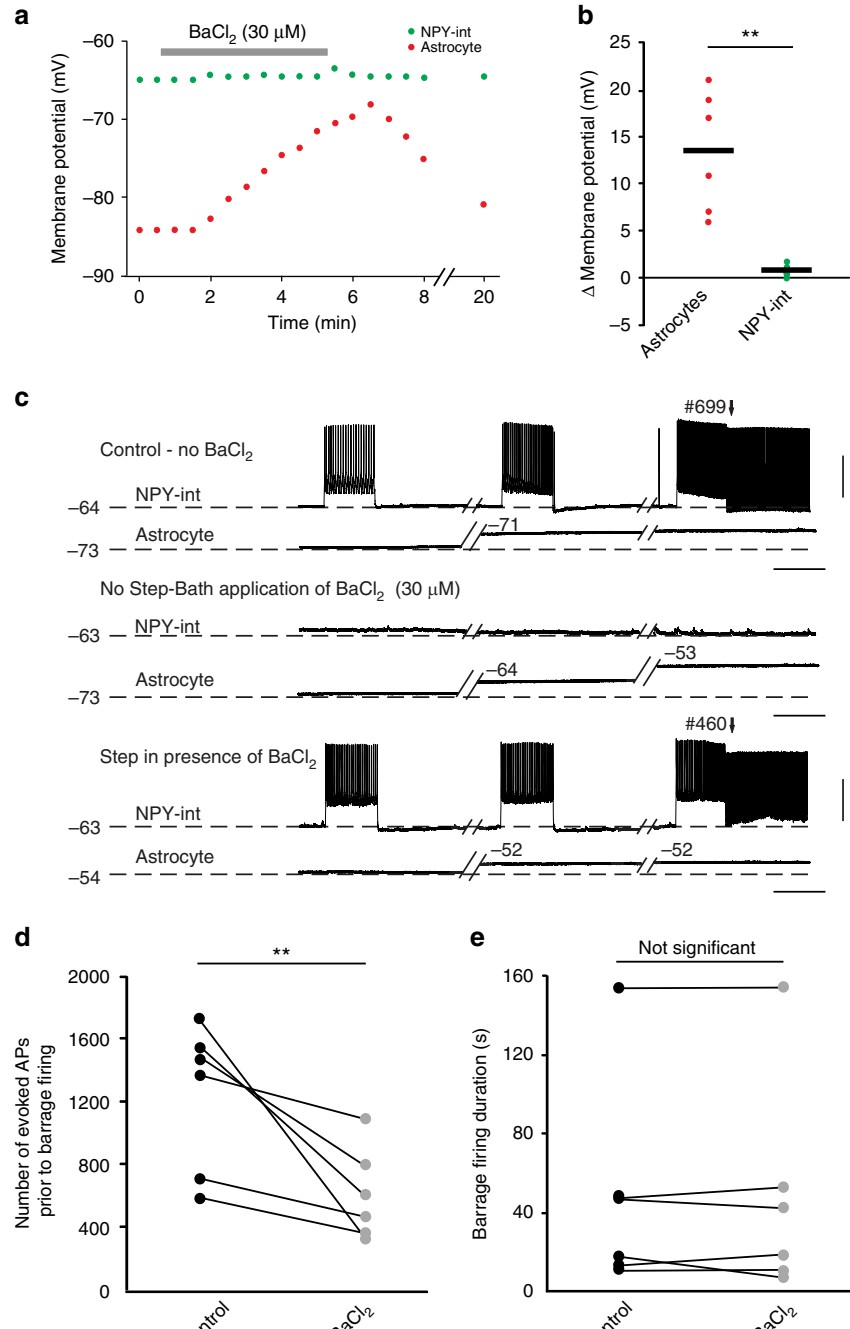

**Fig. 2** Depolarizing astrocytes enhances induction of barrage firing. **a** Effect of bath application of 30 μM BaCl$_2$ on the membrane potential of an NPY interneuron (green) and astrocyte (red) during an example double recording. **b** Change in membrane potential of NPY interneurons and astrocytes in response to bath application of BaCl$_2$ for all recordings ($n = 6$ double recordings from 3 mice; paired $t$ test, **$p = 0.01$). **c** Example recording showing the effect of BaCl$_2$ on astrocyte depolarization, which results in earlier barrage firing generation. Scale bars, 35 mV, 1 s. **d** BaCl$_2$ application resulted in a decrease in the number of action potentials required for barrage firing induction ($n = 6$; paired $t$ test, **$p = 0.002$). **e** BaCl$_2$ application did not affect the duration of barrage firing ($n = 6$). Data are represented as mean ± SEM

in the recorded NPY interneuron (Fig. 4a, b). Notably, chelating calcium in astrocytes did not affect the number of spikes during barrage firing (Fig. 4d, $p = 0.4$). Furthermore, increases in astrocytic calcium signaling always outlasted the duration of barrage firing in the NPY interneuron (Fig. 3h). These findings suggest that, once initiated, barrage firing proceeds and eventually terminates independently of astrocytic calcium signaling, as suggested by astrocytic depolarization described above.

Previous results indicated that L-type voltage-gated calcium channels (L-Ca$_v$) are a possible source of calcium signaling necessary for barrage firing[23]. An additional source of astrocytic calcium could be from intracellular stores. To test this idea, we used cyclopiazonic acid (CPA), an inhibitor of the endoplasmic reticulum Ca$^{2+}$-ATPase. Bath application of 30 μM CPA for 30 min completely blocked barrage firing ($n = 4$, no barrage firing was induced with more than two times the number of evoked action potentials required for

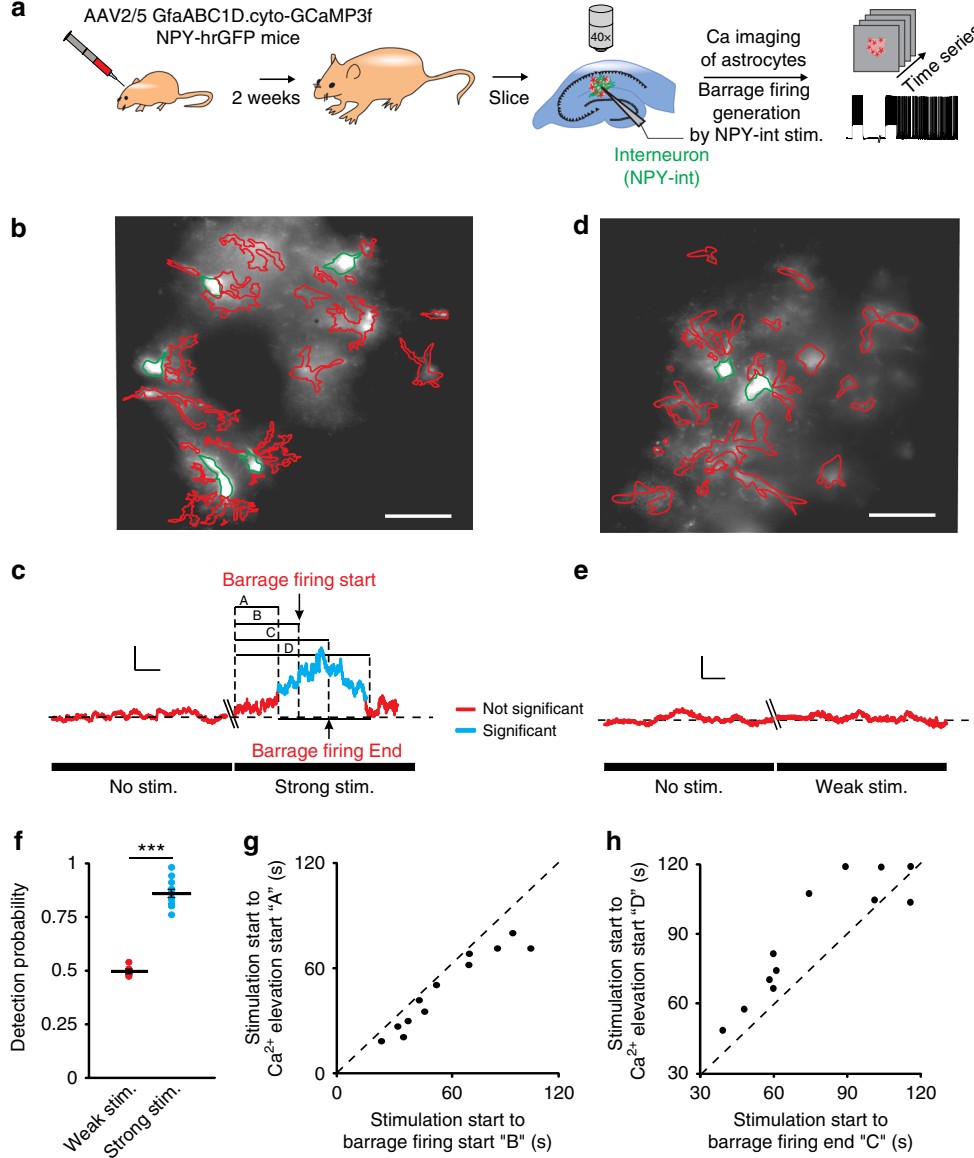

**Fig. 3** Calcium transients increase in astrocytes during barrage firing induction. **a** Schematic demonstrating the experimental method. Author T.D. drew the mice and syringe. **b** Representative image of GFAP-GCaMP3-infected astrocytes illustrating ROI selection. Astrocyte cell bodies outlined in green and their process in red. Scale bar, 40 μm. **c** An increase in astrocytic calcium transients (blue) preceded barrage firing generation in response to strong stimulation (average of all ROIs shown in **b**). Scale bars, 1% ΔF/F, 15 s. Uppercase letters refer to the time from onset of strong stimulation to beginning of detectable of calcium increase (A), barrage firing start (B), barrage firing end (C), and end of detectable calcium increase (D). In this experiment, strong stimulation was used (i.e., leading to barrage firing firing). **d** Representative image of GFAP-GCaMP3-infected astrocytes illustrating ROIs in an experiment with weak stimulation (i.e., not leading to barrage firing. Scale bar, 40 μm. **e** No increase in astrocytic calcium (red) was observed in the absence of stimulation or with weak stimulation, which did not induce barrage firing (average ROIs in **d**). Scale bars, 3% ΔF/F, 15 s. **f** Detection probability for onset of the increase in average calcium transients in astrocytes prior to barrage firing onset was high for trials that resulted in barrage firing induction ($n = 12$ from 8 mice) but not for trials with weak stimuli and therefore no barrage firing (i.e., ~50%—or chance—detection probability; $n = 6$ from 4 mice; ***$p < 0.0001$). **g** Astrocytic calcium increases always preceded barrage firing onset ($n = 12$ from 8 mice). **h** Termination of barrage firing preceded the termination of significant calcium elevation in astrocytes ($n = 11/12$ from 8 mice). Data are represented as mean ± SEM

barrage firing in control) or inhibited its induction ($709 \pm 211$ spikes in control vs. $1307 \pm 212$ spikes after CPA, $n = 7$; paired $t$ test, $p = 0.04$, Supplementary Fig. 3a). Although bath-applied CPA could affect astrocytes, interneurons, or both, the observations that chelating intracellular calcium in astrocytes inhibits barrage firing, whereas chelating intracellular calcium in the NPY interneuron does not[23] suggest that intracellular stores are an important source of the astrocytic calcium increases that are required for the generation of barrage of firing.

**Photostimulation of astrocytes facilitates barrage firing**. To further test the hypothesis that depolarization and calcium signaling in astrocytes are key steps in the induction of barrage firing, we expressed channelrhodopsin (ChR2) in astrocytes using viral transfection and a GFAP promoter (Fig. 5a)[34]. ChR2 was histologically confirmed to be expressed exclusively in astrocytes (Fig. 5b). Activation of ChR2 by low-intensity blue light (2% of the maximum power) caused rapid depolarization of astrocytes by $6.75 \pm 1.25$ mV. Such photostimulation facilitated barrage

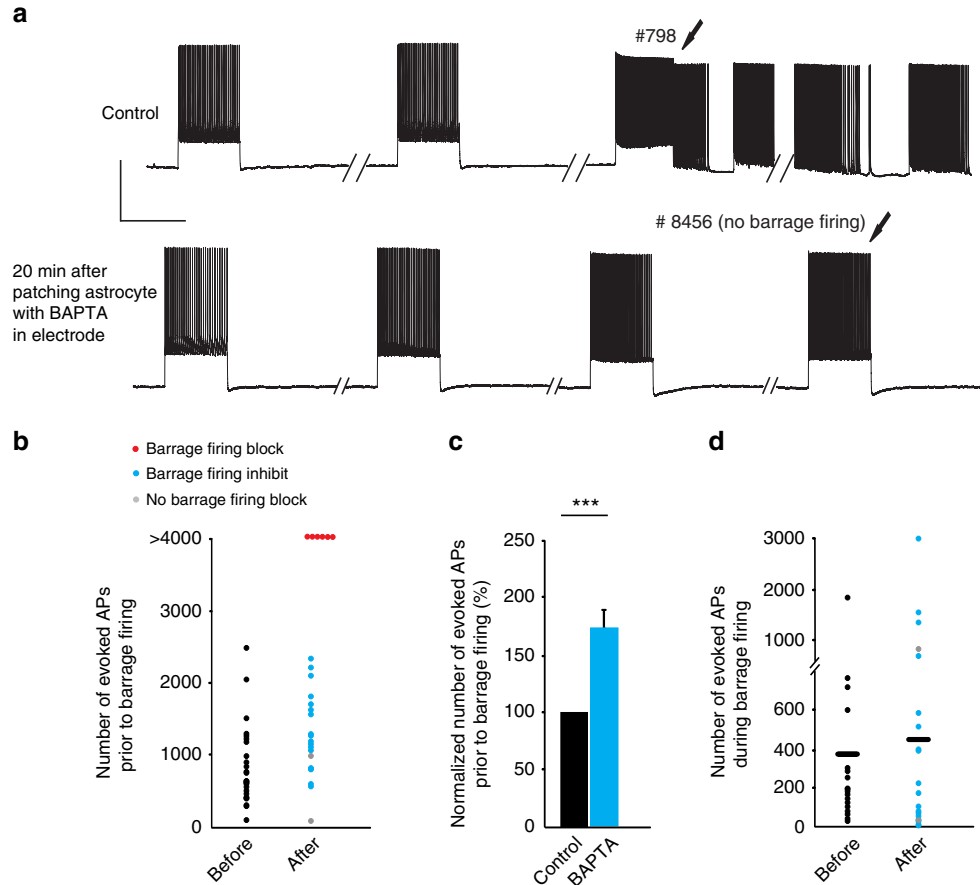

**Fig. 4** Chelating intracellular calcium in astrocytes inhibits barrage firing. **a** Barrage firing induction in an NPY interneuron (control). Barrage firing could not be induced in the same interneuron after chelating $Ca^{2+}$ in the astrocytic network by patching a nearby astrocyte with an electrode containing 50 mM BAPTA. Scale bars, 40 mV, 1 s. **b** The number of evoked spikes to induce barrage firing before (black, $n = 26$ from 20 mice) and after chelating $Ca^{2+}$ in astrocytic network. In some cases, barrage firing was never induced after astrocytic BAPTA (red, $n = 6$ from 6 mice); in some cases, more spikes were required to induce barrage firing (blue, $n = 18$ from 12 mice); in some cases, barrage firing was unaffected (gray, $n = 2$ from 2 mice). **c** The number of spikes required for barrage firing induction increased after astrocytic BAPTA. Normalized number of spikes is calculated only in trials where barrage firing was observed after chelation (blue and gray points in **b**; $n = 20$, paired $t$ test, ***$p = 0.00013$). **d** The number of spikes during barrage firing did not change significantly following astrocytic calcium chelation with intracellular BAPTA (blue and gray points in **b**, $n = 20$). Data are represented as mean ± SEM

firing induction (484 ± 121 spikes during photostimulation vs. 689 ± 167 spikes in control, $n = 5$; paired $t$ test, $p = 0.03$, Supplementary Fig. 4). Increasing blue light intensity to 5–10% of the maximum power resulted in a greater increase in astrocytic depolarization (40.6 ± 4.8 mV, $n = 5$) and an increase in astrocytic intracellular calcium signaling, as monitored using intracellular OGB1 ($\Delta F/F = 2.02 ± 0.4$, $n = 5$; Fig. 5c). The astrocytic $Ca^{2+}$ increase is likely due to opening of voltage-gated calcium channels, as shown previously[22,23]. Although such strong synchronous stimulation of astrocytes is most likely non-physiological, it nevertheless provides a good test of the role for astrocytes in barrage firing generation. ChR2-mediated increases in astrocytic calcium signals in astrocytes occurred after 23.2 ± 8.3 s ($n = 5$), as reported previously[34]. Differences in response to photostimulation across cells may be attributable both to natural biological variability as well as differences in the level of expression of ChR2 and the light intensity used in different trials in our experiments (5–10% max power). Nevertheless, in all cases, combining photostimulation of astrocytes with the barrage-firing induction protocol in NPY interneurons considerably reduced the number of action potentials required to induce barrage firing (271 ± 43 spikes when combined with photostimulation vs. 771 ± 138 spikes in control, $n = 18$; paired

$t$ test, $p = 0.001$, Fig. 5d, e). In some cells ($n = 4/12$), photostimulation alone induced barrage firing in the NPY interneuron (Fig. 5f). Furthermore, ChR2-promoted barrage firing exhibited action potential properties clearly indicative of spike initiation in the distal axon, similar to barrage firing induced by current injections (Fig. 5f). Thus ChR2 activation of astrocytes, through depolarization and enhancement of calcium signaling, results in enhancement of barrage firing induction, further demonstrating a role for astrocytes in triggering barrage firing in NPY interneurons.

**Contributions to barrage firing by GABA and glutamate.** Activation of GABAergic receptors in astrocytes has been shown to elicit $Ca^{2+}$ signals in astrocytes[35–37]. Therefore, we tested the role of GABA receptors in barrage firing. Bath application of the $GABA_A$ antagonist gabazine (SR95531 hydrochloride, 60 μM) did not affect induction of barrage of firing (782 ± 136 spikes required for barrage firing generation in gabazine vs. 721 ± 94 spikes in control, $n = 5$; paired $t$ test, $p = 0.6$; Supplementary Fig. 5a). Similarly, the $GABA_B$ antagonist CGP52432 (10 μM) did not affect the induction of barrage of firing in NPY cells (687 ± 158 spikes required in CGP52432 vs. 679 ± 57 spikes in control, $n = 6$; paired $t$ test, $p = 0.8$, Supplementary Fig. 5b).

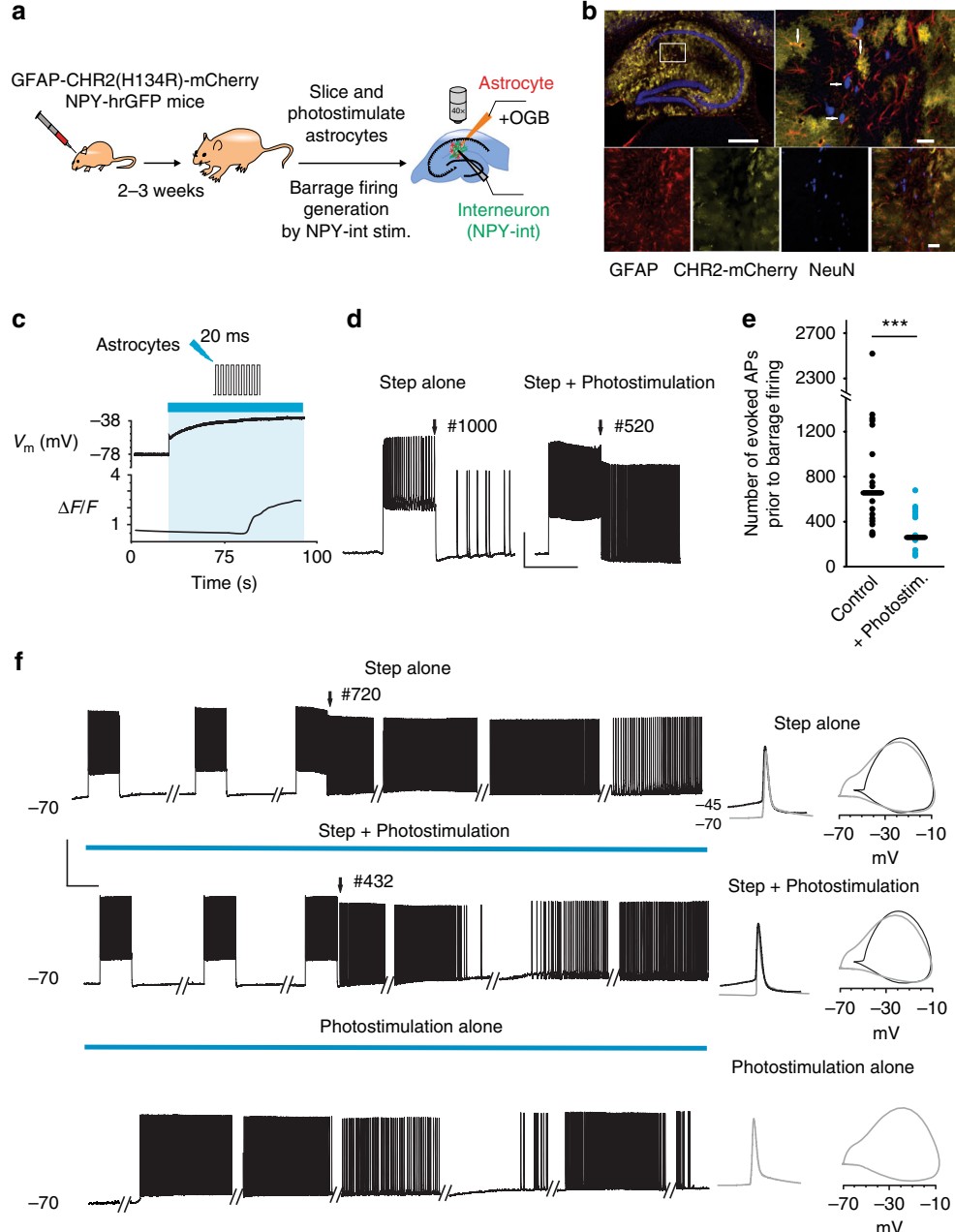

**Fig. 5** Optogenetic activation of astrocytes enhances barrage firing induction. **a** Schematic of experiment protocol. Author T.D. drew the mice and syringe. **b** Image showing co-localization (yellow) of ChR2–mCherry (green) with astrocytes (anti-GFAP, red) but not with neurons (anti-NeuN, blue) in CA1 area. Note that ChR2 is not expressed in all astrocytes (i.e., red instead of yellow). Scale bars, 600 µm top left, 30 µm top right and 40 µm bottom right. **c** Example of recording from a ChR2-mCherry-infected astrocyte and the observed membrane potential depolarization and increase in $Ca^{2+}$ transients in response to photostimulation. **d** Facilitation of barrage firing generation in an example recording with combined step depolarization and photostimulation of ChR2-mCherry-infected astrocytic network (right trace) compared to control condition (i.e., in response to step depolarization alone, left trace). The number of spikes required for barrage firing induction is shown on top of the traces. Scale bars, 30 mV, 1 s. **e** Comparison of the number of evoked action potentials (i.e., step depolarization) required to induce barrage firing with and without photostimulation ($n = 18$ from 10 mice; paired $t$ test, ***$p = 0.001$). **f** Barrage firing induction using three different protocols in the same interneuron. Similar results were observed in three other neurons ($n = 4/12$ from 8 mice; no barrage firing in response to photostimulation alone in $n = 8/12$). Phase plots of barrage firing spikes (right, gray) are similar for all conditions and readily distinguishable from current-evoked spikes (black). Scale bars, 50 mV, 1 s. Data are represented as mean ± SEM

It has also been shown that $Ca^{2+}$ signaling in astrocytes can be mediated by GABA uptake[37]. We focused on a possible role for GAT-3 in generation of barrage firing, since it is the main GABA transporter present in astrocytes throughout the hippocampus formation[38] and it has been shown to evoke $Ca^{2+}$ signaling in astrocytes[37]. Bath application of SNAP 5114 (100 µM), a selective GAT3 inhibitor[39], significantly blocked or inhibited barrage firing

(no barrage firing was induced with more than two times the number of evoked action potentials required for barrage firing in control, $n = 4$; or barrage firing was induced but required more spikes, $964 \pm 64$ spikes required in SNAP 5114 vs. $598 \pm 30$ spikes in control, $n = 9$, paired $t$ test, $p = 0.02$, Supplementary Fig. 5c). These findings suggest that GABA released following stimulation of NPY interneurons could trigger GABA uptake by astrocytes,

thus resulting in increased astrocytic calcium concentration, which is required for initiation of barrage firing.

We next explored two likely paths through which astrocytes could affect NPY interneuron activity, leading to barrage firing. Depolarization of astrocytes results in release of different gliotransmitters—ATP and glutamate—which may in turn affect the NPY cells so as to induce barrage of firing. It has been previously observed that suramin, a non-selective P2 purinergic antagonist, had no effect on barrage of firing (Sheffield, PhD thesis). We further tested whether blocking P1 receptors can interfere with barrage firing induction. Bath application of 100 nM ANR-94 (100 nM), an antagonist of the adenosine receptors[40], had no effect on barrage firing ($532 \pm 97$ spikes required in ANR-94 vs. $502 \pm 58$ spikes in control, $n = 4$; paired $t$ test, $p = 0.6$; Supplementary Fig. 3b). Finally, it has been reported that cultured astrocytes release ATP through pannexin1 hemichannels;[41] however, block of these channels by probenecid (2.5 mM), a pannexin 1 inhibitor, had no effect on the ability of NPY cells to generate barrages of action potentials ($1091 \pm 254$ spikes required for generation of barrage firing in probenecid vs. $1255 \pm 360$ spikes in control, $n = 5$; paired $t$-test, $p = 0.3$; Supplementary Fig. 3c). These findings show that ATP release from astrocytes is most likely not required for the generation of barrage firing.

Previous studies have shown that calcium-dependent release of glutamate from astrocytes could activate neuronal metabotropic glutamate receptors and modulate the activity of interneurons in the hippocampus[42]. Therefore, we tested the effect of blocking mGluRs on induction of barrage of firing. Following bath application (for 30 min) of a cocktail of group I mGluR antagonists (100 µM LY367385 and 50 µM MPEP hydrochloride) and a group II/III mGluR antagonist (200 µM CPPG), induction of barrage of firing was blocked (no barrage firing after >2500 evoked action potentials) in 67% of NPY interneurons (6/9 cells in 3 mice) and was inhibited in the other 33% (3/9 cells; $2083 \pm 597$ spikes required in group I/II/III mGluR antagonists vs. $820 \pm 275$ spikes in control; Supplementary Fig. 5d). Furthermore, when slices were pre-incubated in this cocktail for 15–30 min, barrage firing was absent in 100% of cases ($n = 16/16$ cells in 3 mice). Control recordings from non-incubated slices from the same animals showed barrage firing 83% of the time ($n = 5/6$ cells in 3 mice). Furthermore, bath application of group I mGluR antagonists alone (100 µM LY367385 + 50 µM MPEP; no pre-incubation) was sufficient to inhibit the induction of barrage of firing ($1699 \pm 335$ spikes required after group I mGluR antagonists vs. $544 \pm 86$ spikes in control, $n = 5$; paired $t$ test, $p = 0.03$; Supplementary Fig. 5e).

Finally, we studied the effect of the glutamate transporter blocker L-TBOA (25 µM) on barrage firing. TBOA facilitated the induction of barrage firing by ~30% ($769 \pm 95$ spikes required for generation of barrage firing in TBOA vs. $1175 \pm 118$ spikes in control, $n = 6$; paired $t$ test, $p = 0.02$; Supplementary Fig. 5f), further showing that glutamate participates in the induction of barrage firing. Together these findings indicate that reuptake of GABA and activation of mGluRs are both necessary for induction of barrage of firing in NPY interneurons. As noted above, bath-applied drugs could exert their effects on astrocytes, interneurons, or both, so additional considerations influence interpretation of these results (see Discussion).

## Discussion

Our previous work[22,23] is consistent with the notion that a latent variable represents the state of a leaky integrator that responds to interneuron spiking (Fig. 6a). When this latent variable reaches a threshold, it drives barrage firing, possibly with a delay corresponding to biochemical and physiological processes between this threshold crossing and the manifestation of barrage firing. Here we provide several lines of evidence that this latent variable is instantiated in astrocytic networks. On the basis of our results, we propose the following model that involves both neuron-to-astrocyte and astrocyte-to-neuron signaling. Astrocytes are in close contact with the axons of NPY interneurons. The physical substrate for the astrocyte interneuron interaction is likely between the axons of NPY interneurons, where barrage firing is generated[22], and the fine processes of astrocytes (Fig. 6b), which have been shown to form close contacts with neurons in the hippocampus[43]. Extensive firing of at least one NPY interneuron, over the course of tens of seconds, results in a slow integrative process in the astrocytic network, which is manifested by its depolarization and calcium signaling. These processes ultimately result in barrage firing in the distal axons of one or more NPY interneurons. The axons of multiple interneurons may interact functionally with the same astrocyte and/or network of astrocytes (Fig. 6b), as we have previously shown that evoked firing in one NPY interneuron can lead to barrage firing in another nearby NPY interneuron[22]. Although astrocytic depolarization and calcium signals are related to the initiation of barrage firing in NPY interneurons, the onset of these events precedes and outlasts barrage firing by tens of seconds. Therefore, astrocytic depolarization and astrocytic calcium elevation are early steps in the initiation of barrage firing, but they are not its proximate cause. In support of this model, we present evidence comprising a predictive correlation and a gain-of-function effect associated with astrocytic depolarization, a predictive correlation and an inhibition/loss-of-function effect associated with astrocytic calcium signaling, and a gain-of-function effect of ChR2, which causes both astrocytic depolarization and calcium signaling.

Although barrage firing results from a slow integrative process involving hundreds of evoked action potentials over tens of seconds (usually more than a minute), the relatively abrupt onset of barrage firing suggests that the slow integrative process in astrocytes ultimately reaches a threshold that sets off barrage firing in NPY interneurons. However, the locus and nature of this thresholded event are unclear. One possibility is that this event occurs within the astrocytic network, triggering release of a gliotransmitter that activates receptors on the NPY interneuron, which in turn triggers the barrage firing. Alternatively, glio-transmitter may be gradually released from astrocytes (e.g., in parallel with elevation of intracellular calcium), and the relatively abrupt onset of barrage firing may reflect a threshold that is reached in the neuron in response to long-lasting receptor activation by the gliotransmitter. Gradual changes in extracellular potassium concentration may also contribute to the slow integrative process preceding barrage firing, but direct depolarization of interneuron axons cannot fully explain it, as this mechanism cannot explain the specificity of barrage firing to NPY inter-neurons. In further support of the concept of the specificity of astrocyte–neuron interactions, a recent study showed that separate classes of astrocytes interact preferentially with synapses onto specific neuronal cell types[44].

Although the molecular mechanisms of neuron-to-astrocyte and astrocyte-to-neuron signaling responsible for barrage firing are not yet fully understood, our results provide some clues. Data presented here and elsewhere[22,23] suggest a possible model of the molecular nature of the interactions between NPY interneuron axons and astrocytic processes (Fig. 6c). In this model, evoked firing of the NPY interneuron results in release of GABA and the GAT-3 transporter moves GABA along with $Na^+$ into astrocytes. As suggested previously[37], increased intracellular $Na^+$ inhibits the $Na^+/Ca^{2+}$ pump, resulting in increased intracellular $Ca^{2+}$ and activation of $Ca^{2+}$-mediated $Ca^{2+}$ release from internal sources. Astrocytic depolarization (in part, perhaps, as a result of

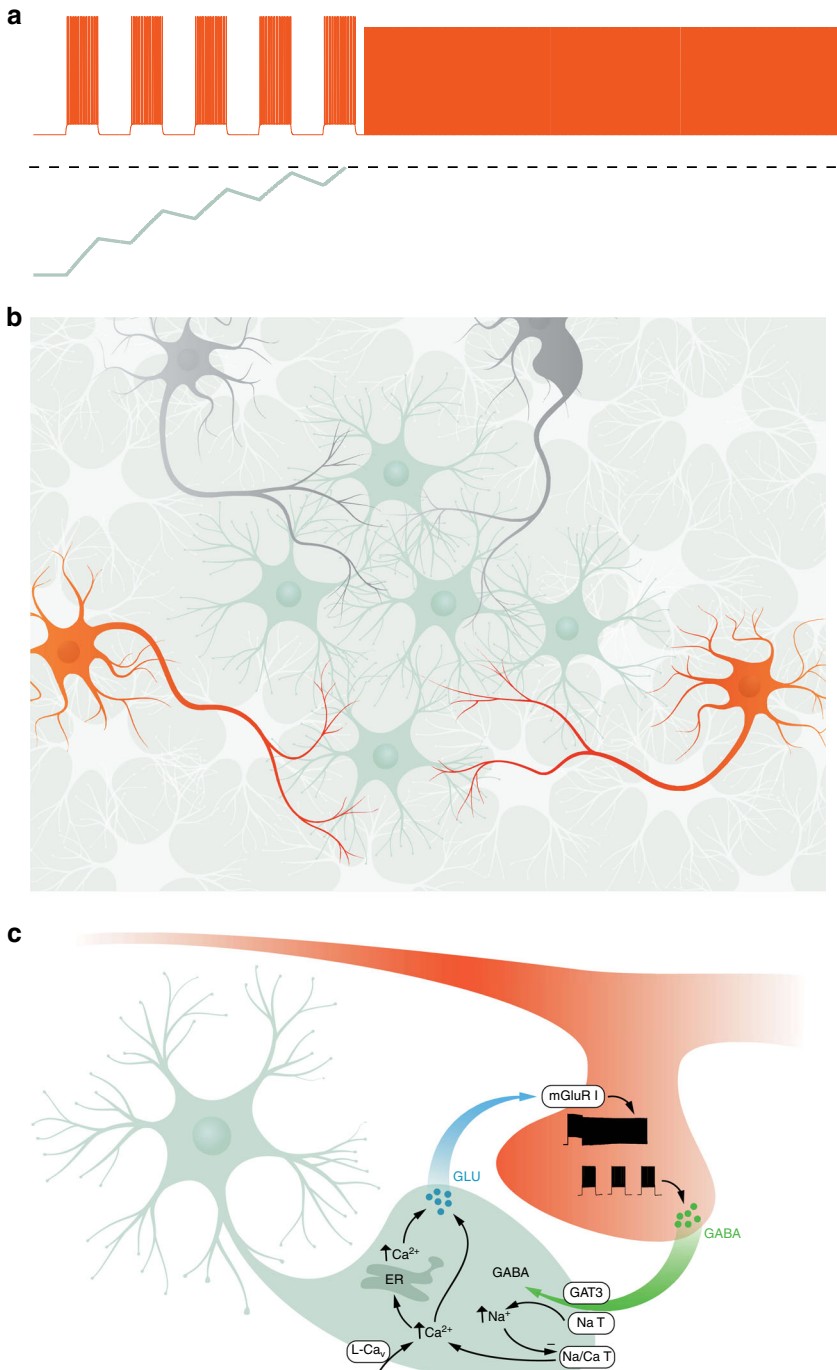

**Fig. 6** Schematic model for the interaction between astrocytes and NPY interneurons to induce barrage firing. **a** Repeated spiking (orange trace) of an NPY interneuron eventually results in barrage firing. As shown schematically, spikes are evoked by 1-s-long step depolarizations; only five steps are shown, though normally many more are required. During barrage firing, spikes emanate from the distal axon, resulting in somatic spikes with no apparent underlying depolarization. Evoked spikes (i.e., during the step depolarizations) lead to step increase of a hidden variable (green trace, corresponding to an unknown biomolecule in the astrocyte) that decays as a leaky integrator, modeled here using a decay time constant of ~7.7 s. When this hidden variable reaches a threshold (dashed line), it results in processes that elicit barrage firing, following a delay, as shown here. Vertical scale is arbitrary. **b** NPY interneurons (orange) and other interneurons (gray) overlap with astrocytes (green and white), but only NPY interneurons are capable of generating barrage firing. The axons of multiple NPY interneurons may interact with the fine processes of many astrocytes, many of which may be interconnected via gap junctions. **c** Schematic showing a model of the potential molecular steps in astrocytes leading to barrage firing in NPY interneurons. See text for details. GLU glutamate, ER endoplasmic reticulum, GAT3 GABA transporter, NaT sodium transporter, Na/Ca T sodium/calcium transporter, mGluR I group I metabotropic glutamate receptor. Illustrations in Fig. 6b, c were drawn by Julia Kuhl

increased extracellular $K^+$) and activation of L-Ca$_v$ channels may also contribute to the initial rise in calcium. The increased $Ca^{2+}$ results in release of glutamate from the astrocyte, which acts on group I mGluR receptors on interneuron axons, which in turn produces barrage firing.

A significant caveat of all of the pharmacological experiments described here is that drugs may act on receptors in astrocytes, NPY interneurons, or both, as well as in other cells in the slice. Thus further testing of the molecular hypothesis described here (Fig. 6c) will require cell-type-specific manipulations. Furthermore, the molecular players should ideally be localized to sites of interaction between fine astrocytic processes and the axons of NPY interneurons using methods for high-resolution imaging. Finally, astrocytes are also connected via gap junctions to oligodendrocytes[45], so we cannot rule out the potential role of oligodendrocytes in the induction of barrage firing. The model presented here will almost certainly require revision as these and other new data become available. The model is, however, consistent with available data and it serves as a useful framework for designing future experiments.

Regardless of the molecular steps involved, our results identify a new role for astrocytes. Although there is extensive evidence that astrocytes can modulate neuronal firing through effects on excitability, synaptic transmission, and synaptic plasticity (see Introduction), we find here that astrocytes can directly drive action potential firing in a specific interneuron subtype. Barrage firing is not a modification of an otherwise normal firing pattern; rather, it involves generation of action potential firing via bidirectional interactions between astrocytes and the distal axons of NPY interneurons. Thus our data support a new role for astrocytes that involves detecting, integrating, and driving action potential firing.

Sophisticated nervous systems must go beyond simple stimulus–response functions in order to incorporate the past into neural computations resulting in present action (behavior). Synaptic plasticity is one mechanism by which this is thought to occur. Another is through slow integration of neural inputs to produce lasting outputs such as persistent firing, which is crucial for diverse processes such as motor control, encoding of head direction, and decision making[21,46]. Slow integration is normally attributed exclusively to neuronal, cell-autonomous processes, and/or reverberation of activity in neuronal circuits[21,47,48]. Here we show that astrocytes are also capable of performing these functions, thus suggesting that they might contribute to a key computational property of complex nervous systems. Moreover, the possibility that astrocytes drive action potential firing in one neuron as a result of detecting activity in other neurons[22] positions them not only as temporal integrators of neuronal activity but also as conduits for the transmission of information between neurons in circuits.

Various functions have been proposed for barrage firing[22,25,26]. First, persistent firing in interneurons occurs in the beta and gamma frequency range, which are believed to play a role in synchronization of principal neuronal activity that could be involved in cognitive processing and psychiatric disorders. Second, barrage firing without any ongoing stimulation, similar to other forms of persistent firing, could be an underlying mechanism for short-term storage of information such as working memory. Finally, a possible function of astrocyte-mediated barrage firing pertains to the fact that recurrent excitatory networks are prone to runaway excitation (epilepsy) when the "loop gain" exceeds unity. An ideal system for aborting epilepsy would first detect when an abnormally large number of neuronal spikes occur and then trigger spatially and temporally extensive inhibition to quench the excitation. Astrocyte-mediated barrage firing has these properties. Notably, under physiological conditions, this

effect could be mediated by activity in a distributed interneuron network, rather than the multiple long stimuli of a single interneuron used in our experiments. The fact that barrage firing occurs exclusively in NPY interneurons is also notable, as NPY has been identified as an endogenous modulator of epileptic activity[49].

The mechanistic insight offered here presents new opportunities for future studies of the function of barrage firing of interneurons in circuit function, cognition, and behavior. Newly available tools to manipulate astrocytes in vitro and in vivo[50–52] may further facilitate future efforts to identify the detailed molecular mechanisms mediating signaling from neurons to astrocytes and back, as well as the activity patterns and behavioral conditions that may engage these mechanisms to influence perception, cognition, and behavior[2,3,7]. Combined with the fact that the number of astrocytes rivals the number of neurons in the brain, especially in the computationally sophisticated hippocampus and cerebral cortex, our results suggest that astrocytic input–output properties will eventually need to be included in cellular-scale models of nervous system function[2,4,31].

## Methods

**Transgenic mouse strains**. All procedures were performed in accordance with protocols approved by the Janelia Institutional Animal Care and Use Committee.

Mice from a line expressing humanized *Renilla* GFP under the NPY promoter (NPY-GFP, B6.FVB-Tg, The Jackson Laboratory stock #006417[53]) were used for all experiments in order to visualize NPY-expressing interneurons. In one set of experiments, NPY-GFP were crossed with Cx30 KO or Cx43flox×GFAP-Cre or Cx43flox/Cx30 KO×GFAP-Cre mice to achieve a combination of connexin knockout(s) and GFP labeling of NPY interneurons (Supplementary Fig. 6). Individual KO animals were provided by Dr. Ken D. McCarthy (University of North Carolina). In total, this study is based on data from 111 mice (male and female). No statistical methods were used to predetermine sample size. The experiments were not randomized. The investigators were not blinded to allocation during experiments and outcome assessment.

**Surgery and in vivo microinjections of adeno-associated virus (AAV) 2/5**. Mice at postnatal days 15–17 were anesthetized using continuous isoflurane (induction at 5%, maintenance at 1–2.5% vol/vol) delivered through a nose pit. During the surgery, depth of anesthesia was monitored and adjusted as needed. Once fixed in the stereotaxic apparatus (David Kopf Instruments), craniotomies of 2–3 mm were made over the left parietal cortex and 100 nl of virus (rAAV2-retro[54] GFAP-ChR2(H134R)mCherry or AAV2/5 gfaABC1D GCaMP3) was unilaterally injected using a pump (Pump11 PicoPlus Elite, Harvard Apparatus) attached to glass pipettes (1B100-4, World Precision Instruments) positioned at a predetermined injection site (2.3 mm posterior to bregma; 1.8 mm lateral to midline; 1.2 mm from the pial surface). The wound was sutured afterward and mice were used for in vitro calcium imaging and electrophysiology after 16–18 days to allow for sufficient transgene expression.

**Hippocampal slice preparation**. Parasagittal slices of hippocampus (300 μm) were made from both male and female mice (P16–P28 as detailed previously[23]). Mice were deeply anesthetized with isoflurane (unresponsive to hind paw pinch) and decapitated. The brains were quickly removed and placed in ice-cold artificial cerebrospinal fluid (ACSF; composition in mM: 125 NaCl, 2.5 KCl, 25 NaHCO$_3$, 1.25 NaH$_2$PO$_4$, 1 MgCl$_2$, 2 CaCl$_2$, 25 dextrose) superfused with carbogen (5% CO$_2$–95% O$_2$) for 2 min before slicing. A similar ACSF solution was used for incubating slices during all recordings. The hippocampus was sectioned in the parasagittal plane with a vibrating microtome filled with carbogenated chilled ACSF. Next, slices were warmed up to 30 °C in an incubation chamber with bubbled ACSF. In order to visualize astrocytes for patch-clamp recording, the slices were first incubated in warm ACSF (30 °C) containing 1 μM sulphorhodamin-101 (SR101) for 20 min and then transferred and maintained in the regular ACSF at room temperature until placed in the recording chamber.

**Hippocampal slice electrophysiology**. For recording, slices were transferred to the chamber where they were maintained by constant perfusion (2–3 ml/min) of carbogenated ACSF at 30–36 °C. Patch-clamp electrodes were pulled from borosilicate glass (5–8 MΩ tip resistance) and filled with intracellular solution containing 135 mM potassium gluconate, 7.5 mM KCl, 10 mM sodium phosphocreatine, 10 mM HEPES, 2 mM MgATP, 0.3 mM NaGTP, and 0.5% mM biocytin. Recordings were made using a Dagan BVC-700A amplifier (Dagan Corporation, Minneapolis, MN). The stability of the recording was checked by periodically adjusting capacitance compensation and bridge balance while

observing voltage responses to 50 pA current injections. Somatic whole-cell patch clamp recordings were made using one or two amplifiers (BVC-700, Dagan). Electrophysiological data were digitized with an ITC-16 A/D board (Heka Electronik) under control of custom software (DataPro, https://github.com/adamltaylor/DataPro) programmed in IGOR Pro (WaveMetrics). Recordings from NPY interneurons at the border of SLM and SR were performed in current-clamp mode, maintaining the membrane potential between –65 and –70 mV (injection of 0–100 pA). To induce barrage firing, 1-s current injections were delivered at the beginning of 3 s sweeps. The amplitude of these current steps was set so as to generate ~40 spikes during the 1-s step (180–700 pA, depending on membrane properties such as resistance, adaptation, etc.). Once barrage firing was initiated, as evidenced by firing rate of >4 Hz for at least 1 s after the step current injection ended, further current injections were stopped and barrage firing was recorded for up to 4 min. Only NPY interneurons that were able to generate barrage firing in response to <2000 evoked spikes (in control) were considered as barrage firing cells; other cells were excluded from further analysis. A 5-min recovery interval was included between successive barrage firing inductions in the same neuron. A maximum of 5 epochs of barrage firing were induced in each interneuron and no significant difference was observed in the number of spikes required for barrage firing induction between the 3 different trials for each interneuron (<1% change between trials). Astrocytes were distinguished from interneurons based on their morphology (i.e., smaller somata, with a diameter of <10 μm) and electrophysiological properties (i.e., more hyperpolarized resting membrane potential, usually <−75 mV) and low initial input resistance of <100 MΩ). Some of the astrocyte recordings were confirmed later by biocytin staining; patch-clamp recording from a single astrocyte typically labeled a large network of astrocytes (58 ± 8 cells, $n = 10$ slices), as reported previously[27,55].

**Intracellular calcium imaging**. Calcium signals were monitored by fluorescence microscopy from astrocytes in SLM and SR layers of CA1 region of the hippocampus; signals were monitored either with the genetically encoded calcium sensor GCaMP3 or with the calcium-sensitive dye Oregon green BAPTA-1 (OGB-1). A blue LED and a 470 nm bandpass filter (Chroma ET470/4×) was used to illuminate the slice. The emitted green light was detected by a 525 nm bandpass emission filter (Chroma, ET525/50 m). A CCD camera (Rolera-XR Fast 1394, QImaging) mounted to a microscope with a 40×/0.8 NA water-immersion objective (Leica Microsystems) was used to collect fluorescence at a frame rate of 7 Hz. Custom software (GECIquant)[56] was used to detect and analyze $Ca^{2+}$ signals in a semi-automatic manner. After thresholding in ImageJ software, a polygon was drawn manually around the astrocyte area of interest. Somatic $Ca^{2+}$ fluctuations were identified by searching for areas >30 μm² and later visually inspected to mark the anatomically well-defined cell body and initial proximal segments of processes of astrocytes ("somatic" region of interest (ROI)). In order to detect $Ca^{2+}$ fluctuations that were observed in astrocyte processes, somata were first demarcated and masked in order to be able to accurately threshold astrocyte processes. Software parameters were then set for "calcium waves" or "expanding signal." In brief, such local waves were present in astrocyte processes and appeared as expansions and contractions that spread between adjacent pixels over an area of >5 μm². All areas were visually inspected after automated detection. We did not consider calcium microdomains with areas <4 μm². For each ROI, basal fluorescence intensity ($F$) was determined during 100 s periods with no stimulation; fluorescence changes ($\Delta F$) were normalized to the basal intensity ($\Delta F/F$). Note that because of our resolution of the imaging process, small changes in some of the signals in finer processes might have been missed here, but this would not affect our conclusions. For detecting astrocytic $Ca^{2+}$ signals evoked by photostimulation, ChR2–mCherry-expressing astrocytes were bulk-loaded with OGB-1 (200 μM) through a patch pipette. $F$ was determined during 100 s periods with no photostimulation and changes after blue LED illumination were normalized to the basal intensity ($\Delta F/F$).

**Optogenetic stimulation**. ChR2-expressing astrocytes were identified by visualizing co-expression of mCherry. Blue-light pulses (20/20 ms duty cycle for 180 s) were used for ChR2 activation and astrocyte depolarization. In a subset of experiments, astrocytes were filled with OGB-1 to monitor $Ca^{2+}$ responses associated with photostimulation of ChR2 (Fig. 5b).

**Data analysis and statistics**. All data analysis was performed in Matlab (The MathWorks, Nattick, MA, USA) using custom-written routines. Barrage firing duration and number of spikes during it normally distributed between cells is as previously reported. Pooled data from multiple cells were tested for significant differences using either repeated-measures ANOVA or paired or unpaired two-tailed Student's $t$ test comparisons. For all statistical tests, the significance level is considered at 0.05. All measurements are presented as mean ± SEM unless otherwise indicated. The numbers of experiments noted in figure legends reflect independent experiments performed on different days.

**Feature detection analysis**. To distinguish periods of significant depolarization or calcium increases in astrocytes (e.g., following step current injections of interneurons to induce barrage firing) from the baseline fluctuations, a receiver

operating characteristic (ROC) curve[57–59] was used to find the detection probability (i.e., area under the curve) of a significant difference across trials. We compared baseline values (30 s for membrane potentials and 120 s for calcium signals) to those obtained during each 3-s sweep that contained depolarizing steps to induce barrage firing or during barrage firing once initiated (a total of 40 sweeps). Matlab was used to calculate probability estimates from a logistic regression model fit to the control and step distributions and obtain false-positive and true-positive values of the ROC curve for each sweep. The area under the curve indicates the ability of the classifier to detect deviation from baseline (i.e., detection probability): an area of 1.0 indicates perfect classification; an area of 0.5 indicates chance discrimination by the classifier. As a measure of the dynamics of the change in astrocyte membrane potential, the time required for depolarization between 20% and 80% of the maximum value was calculated.

**Code availability**. Custom software used for this project is accessible through Figshare (https://doi.org/10.6084/m9.figshare.6991115), also accessible via our laboratory website (http://www.janelia.org/lab/spruston-lab/resources).

**Data availability**
The raw data for all figures are available at Figshare (https://doi.org/10.6084/m9.figshare.7098665), also accessible via our laboratory website (https://www.janelia.org/lab/spruston-lab/resources).

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

## Acknowledgements
We thank Julia Bachman, Brett Mensh, Gerry Rubin, Bernardo Sabatini, David Stern, Karel Svoboda, and Johan Winnubst for helpful comments on the manuscript and/or the Abstract and members of the Spruston lab for valuable discussions. We also thank Kim Ritola and Sarada Viswanathan for viruses, Jared Rouchard for surgeries, Brenda Shields for histology, Ken McCarthy (University of North Carolina) for Cx43 flox and Cx30 KO mice, Deanna Otstot and Amanda Zeladonis for animal care and breeding, Julia Kuhl for artwork, and Baljit Khakh for assistance with reagents and imaging analysis software. This work was supported by the Howard Hughes Medical Institute.

## Author contributions
T.D. and N.S. conceived the project, designed the experiments, and wrote the manuscript. T.D. performed experiments and analyzed the data. J.L. collected and analyzed data for Supplementary Figure 2.

## Additional information

**Competing interests:** The authors declare no competing interests.

