## [Peer Review File · Nature Communications]

Editorial Note: this manuscript has been previously reviewed at another journal that is not operating a transparent peer review scheme. This document only contains reviewer comments and rebuttal letters for versions considered at *Nature Communications*. Mentions of prior referee reports have been redacted.

REVIEWERS' COMMENTS:

Reviewer #1 (Remarks to the Author):

This is a revised version of a previously submitted manuscript that follows up on previous findings from this team showing that repeated depolarization of a subtype of genetically identified hippocampal interneuron (NPY interneurons) can initiate, in the distal axon, long-lasting barrages of action potentials (termed barrage firing: BF) that outlast stimulation by tens of seconds. The main claim of this paper is that astrocytes integrate neural activity temporally and in return drive this barrage firing. This claim is supported by the facts that: 1) BF is associated to astrocytic depolarization and rise of intracellular Ca²⁺ and 2) that stimulation (with Channelrhodopsin) or inhibition (with dialysis of the astrocytic network with BAPTA) of astrocytic activity facilitates or inhibits BF respectively. Newly added experiments since the first submission suggest that activation of astrocytes may depend on GAT3 (astrocytic GABA transporter) which may lead to increased intracellular Ca²⁺ (after release from internal stores), release of glutamate and activation of group I mGluR receptors on NPY interneurons causing BF. These results are interesting and provide additional evidence for the importance of astrocytes in processing of nervous information. This is a considerably strengthened manuscript. The authors should be congratulated on their efforts to address the reviewers concerns. The added results provide more convincing evidence supporting their hypothesis and offer insights into potential mechanisms involved in this neuron-astrocyte-neuron interaction. Some of these experiments would have provided even stronger arguments in favor of astrocytic implication if astrocyte specific drugs would have been used (to block glutamate transporter for instance), and it would have been nice to test whether local application of group I mGluR receptors agonist can restore BF inhibited by bath application of SNP (or CPA or BAPTA in the astrocytes). Nevertheless, the results support the main claim of the paper that astrocytes “participate” to generation of BF. This being said, the authors should still be cautious with some terms that they use since they have never been able to either completely block BF by inhibiting astrocytes or to trigger BF by stimulating astrocytes. They have obtained these effects in some cases, but in most cases they get enhancement or inhibition of BF. They do, however, provide good reasons and explanations why this may be difficult to achieve in all cases.

Minor suggestions and/or corrections:

The title is catchy but very vague and does not provide any information about the phenomenon described in the paper. It may be mistaken for a review article. Consider revising.

Introduction

1st paragraphe, line 6: remove “d” to “and” before immune

1st paragraphe, line 14: remove “,” before reference 9 (after synapse)

Results

Section on astrocytic depolarization and barrage firing, 2nd line: The authors should mention that they use SR101 to record from identified astrocytes (on the basis of their methods). It would increase trust from the fact that they recorded from astrocytes indeed because the other criteria given (morphology, RMP and absence of spikes) are not sufficient to distinguish astrocytes from other glial cells.

Same section, 1st paragraph on page 3: Reference to fig 1h should be on line 12 (just before the fifth argument), and the figure for that fifth argument should be 1j. Also the text related to that fifth argument states "Fifth, in the majority of recordings (11 out of 14) astrocyte depolarization outlasted barrage firing in the NPY interneuron". The figure (1j) shows only 12 data point (not 14) of which astrocytic depolarization outlasted BF in only 3 cases, was equal to BF in 4 and shorter than BF in 4 cases. Thus, the sentence should be rewritten.

Page 3, 2nd paragraph: Authors cannot claim that recorded cells were astrocytes just because biocytin intrapipette spread to other coupled cells. In many brain areas (including the hippocampus), astrocytes also couple to oligodendrocytes, and it is not unlikely that this coupling may play a role in BF. It would have actually been nice to have this considered in the discussion.

Section on astrocytic calcium and barrage firing: It would have been nice to see individual astrocytic responses to BF, rather than an average trace. It would have provided a nice appreciation of variability in timing. This comment was also raised by another reviewer and the authors provided a response to it, but still....

Figure 3: It is confusing that the x and y axes of panels g and h are flipped.

Extended data Figure 2: Panel A. Is the time scale different from one trace to the other ? I don't think so, but if yes add a scale. If not, why is the BF lasting 9.3 sec of the 4ht trace shorter un duration than the one of the 5th trace (7.8 s) ?

Reviewer #2 (Remarks to the Author):

The authors have properly addressed most of my concerns [redacted]. They have thoroughly revised their manuscript by adding a number of important experiments, and by substantially modifying the main text. Their study provides good evidence that NPY interneurons and astrocytes interact functionally in acute hippocampal slices. As acknowledged by the authors, the physiological relevance of their findings remains to be tested in vivo. Although several loose ends remain regarding the molecular mechanism underlying barrage firing, they did a serious effort to provide some clues. At this point I only have two minor comments:

- Readers would like to see some explanation for the NPY interneuron specificity, that is, what accounts for the fact that these interneurons (but not others) undergo barrage firing? A few lines in discussion will do it.

- What is the role of group II/III, if any, in barrage firing? To directly address this point, the authors may want to use selective antagonists for these receptors.

Responses to reviewers

Reviewer 1

This is a revised version of a previously submitted manuscript that follows up on previous findings from this team showing that repeated depolarization of a subtype of genetically identified hippocampal interneuron (NPY interneurons) can initiate, in the distal axon, long-lasting barrages of action potentials (termed barrage firing: BF) that outlast stimulation by tens of seconds. The main claim of this paper is that astrocytes integrate neural activity temporally and in return drive this barrage firing. This claim is supported by the facts that: 1) BF is associated to astrocytic depolarization and rise of intracellular Ca²⁺ and 2) that stimulation (with Channelrhodopsin) or inhibition (with dialysis of the astrocytic network with BAPTA) of astrocytic activity facilitates or inhibits BF respectively. Newly added experiments since the first submission suggest that activation of astrocytes may depend on GAT3 (astrocytic GABA transporter) which may lead to increased intracellular Ca²⁺ (after release from internal stores), release of glutamate and activation of group I mGluR receptors on NPY interneurons causing BF. These results are interesting and provide additional evidence for the importance of astrocytes in processing of nervous information.

This is a considerably strengthened manuscript. The authors should be congratulated on their efforts to address the reviewers concerns. The added results provide more convincing evidence supporting their hypothesis and offer insights into potential mechanisms involved in this neuron-astrocyte-neuron interaction. Some of these experiments would have provided even stronger arguments in favor of astrocytic implication if astrocyte specific drugs would have been used (to block glutamate transporter for instance), and it would have been nice to test whether local application of group I mGluR receptors agonist can restore BF inhibited by bath application of SNP (or CPA or BAPTA in the astrocytes).

We thank the reviewer for positive comments and suggestions. We hope to perform additional pharmacological studies in the future, combining it with other strategies to manipulate receptor expression in astrocytes and/or interneurons. Note: because BF is generated far from the soma, in the distal axon, which branches extensively, localized drug application would be extremely difficult.

Nevertheless, the results support the main claim of the paper that astrocytes “participate” to generation of BF. This being said, the authors should still be cautious with some terms that they use since they have never been able to either completely block BF by inhibiting astrocytes or to trigger BF by stimulating astrocytes. They have obtained these effects in some cases, but in most cases they get enhancement or inhibition of BF. They do, however, provide good reasons and explanations why this may be difficult to achieve in all cases.

We agree with the reviewer and have now tried to further tone down the wording in the text.

Minor suggestions and/or corrections:

The title is catchy but very vague and does not provide any information about the phenomenon described in the paper. It may be mistaken for a review article. Consider revising.

We have revised the title to address the comment.

Introduction

1st paragraphe, line 6: remove “d” to “and” before immune

Corrected.

1st paragraphe, line 14: remove “,” before reference 9 (after synapse)

All quotation marks in the text are removed.

Results

Section on astrocytic depolarization and barrage firing, 2nd line: The authors should mention that they use SR101 to record from identified astrocytes (on the basis of their methods). It would increase trust from the fact that they recorded from astrocytes indeed because the other criteria given (morphology, RMP and absence of spikes) are not sufficient to distinguish astrocytes from other glial cells.

Thank you. It is added.

Same section, 1st paragraph on page 3: Reference to fig 1h should be on line 12 (just before the fifth argument), and the figure for that fifth argument should be 1j. Also the text related to that fifth argument states “Fifth, in the majority of recordings (11 out of 14) astrocyte depolarization outlasted barrage firing in the NPY interneuron”. The figure (1j) shows only 12 data point (not 14) of which astrocytic depolarization outlasted BF in only 3 cases, was equal to BF in 4 and shorter than BF in 4 cases. Thus, the sentence should be rewritten.

Figures 1i and 1j display data from the same set of experiments. Two of the points were inadvertently left out of Fig. 1j. It is now fixed. Also, references to figures and number of astrocytes outlasting BF (n=4) is now corrected. As the reviewer points out, it is not the majority and is corrected in the text.

Page 3, 2nd paragraph: Authors cannot claim that recorded cells were astrocytes just because biocytin intrapipette spread to other coupled cells. In many brain areas (including the hippocampus), astrocytes also couple to oligodendrocytes, and it is not unlikely that this coupling may play a role in BF. It would have actually been nice to have this considered in the discussion.

This is an interesting point and we have now added it to the discussion.

Section on astrocytic calcium and barrage firing: It would have been nice to see individual astrocytic responses to BF, rather than an average trace. It would have provided a nice appreciation of variability in timing. This comment was also raised by another reviewer and the authors provided a response to it, but still....

We added examples of individual astrocytic responses to BF and their average trace in two experiments into a new supplementary figure.

Figure 3: It is confusing that the x and y axes of panels g and h are flipped.

This is now corrected.

Extended data Figure 2: Panel A. Is the time scale different from one trace to the other? I don't think so, but if yes add a scale. If not, why is the BF lasting 9.3 sec of the 4ht trace shorter un duration than the one of the 5th trace (7.8 s)?

You are absolutely correct. There is no reason for differences in length of BF shown for different trials (the time scale is cut out). The figure is revised now, showing the same length for all trials and the duration of BF is written in front of the trace and explained in the caption.

Reviewer 2

The authors have properly addressed most of my concerns [redacted]. They have thoroughly revised their manuscript by adding a number of important experiments, and by substantially modifying the main text. Their study provides good evidence that NPY interneurons and astrocytes interact functionally in acute hippocampal slices. As acknowledged by the authors, the physiological relevance of their findings remains to be tested in vivo. Although several loose ends remain regarding the molecular mechanism underlying barrage firing, they did a serious effort to provide some clues. At this point I only have two minor comments:

- Readers would like to see some explanation for the NPY interneuron specificity, that is, what accounts for the fact that these interneurons (but not others) undergo barrage firing? A few lines in discussion will do it.

We now address this point in discussion.

- What is the role of group II/III, if any, in barrage firing? To directly address this point, the authors may want to use selective antagonists for these receptors.

We hope to address this point in a future study. Please also see our response to a similar comment by Reviewer #1.